# Constitutive activation of kappa opioid receptors at ventral tegmental area inhibitory synapses following acute stress

Abigail M Polter[1†], Kelsey Barcomb[1], Rudy W Chen[1], Paige M Dingess[2,3], Nicholas M Graziane[1‡], Travis E Brown[2,3], Julie A Kauer[1*]

[1]Department of Molecular Pharmacology, Physiology and Biotechnology, Brown University, Providence, United States; [2]Neuroscience Program, University of Wyoming, Laramie, United States; [3]University of Wyoming, School of Pharmacy, Laramie, United States

**\*For correspondence:**
Julie_Kauer@Brown.edu

**Present address:** [†]Department of Pharmacology and Physiology, The George Washington University School of Medicine and Health Science, Washington, United States; [‡]Department of Neuroscience, University of Pittsburgh, Pittsburgh, United States

**Abstract** Stressful experiences potently activate kappa opioid receptors (κORs). κORs in the ventral tegmental area regulate multiple aspects of dopaminergic and non-dopaminergic cell function. Here we show that at GABAergic synapses on rat VTA dopamine neurons, a single exposure to a brief cold-water swim stress induces prolonged activation of κORs. This is mediated by activation of the receptor during the stressor followed by a persistent, ligand-independent constitutive activation of the κOR itself. This lasting change in function is not seen at κORs at neighboring excitatory synapses, suggesting distinct time courses and mechanisms of regulation of different subsets of κORs. We also provide evidence that constitutive activity of κORs governs the prolonged reinstatement to cocaine-seeking observed after cold water swim stress. Together, our studies indicate that stress-induced constitutive activation is a novel mechanism of κOR regulation that plays a critical role in reinstatement of drug seeking.

## Introduction

Stress has long been known to be a precipitating factor for the abuse of addictive drugs. Animal models have shown that acute and repeated stressors can escalate intake of addictive substances (*Piazza et al., 1990*; *Ramsey and Van Ree, 1993*; *Goeders and Guerin, 1994*; *Shaham and Stewart, 1994*; *Haney et al., 1995*), and that acute stress can reinstate drug seeking in animals that have undergone extinction training (*Shaham et al., 1994*, *1995*; *Conrad et al., 2010*; *Mantsch et al., 2016*). In recent years, dopaminergic neurons of the VTA have emerged as a significant locus for the overlapping effects of drugs of abuse and stress (*Polter and Kauer, 2014*). Synaptic inputs, by shaping the activity of these neurons, are poised to play an important role in drug seeking. Both acute stress and exposure to drugs of abuse induce a concomitant potentiation of excitatory synapses and loss of long term potentiation of inhibitory synapses (*Ungless et al., 2001*; *Saal et al., 2003*; *Kauer and Malenka, 2007*; *Nugent et al., 2007*; *Chen et al., 2008*; *Niehaus et al., 2010*; *Polter and Kauer, 2014*). Understanding how these synapses are altered by stress will provide key insights into stress-induced drug seeking and provide targets for treating substance use disorders.

A major mediator of stress-induced changes in inhibitory VTA synapses is the dynorphin/kappa opioid receptor (κOR) system. κORs, and their endogenous ligand, dynorphin, are found throughout the brain and have been highly associated with stressful, aversive, and dysphoric experiences (*Bruchas et al., 2010*; *Wee and Koob, 2010*; *Van't Veer and Carlezon, 2013*; *Crowley and Kash, 2015*). Within the VTA, κORs have a range of physiological effects. κORs decrease the firing rate of dopamine neurons through activation of GIRK channels (*Margolis et al., 2003*, *2006*), inhibit

**eLife digest** People who are recovering from drug addiction are more vulnerable to cravings and relapse when under stress. This ability of stress to boost drug relapse can also be shown in animals previously exposed to addictive drugs. Rats can learn to press a lever to administer themselves a dose of cocaine and, during withdrawal, rats previously exposed to the drug will press the lever more often if they are stressed. Indeed, just a few minutes of stress is enough to increase lever pressing for several days.

Stress and addictive drugs both act on a region of the brain called the ventral tegmental area, or VTA, which is part of the brain's reward system. Stress indirectly increases the activity of the VTA. It does so by activating a protein on the surface of VTA neurons called the kappa opioid receptor (κOR for short). Previous studies revealed that five minutes of stress increases the activity of κORs in the VTA of rats for five days. Conversely, blocking κORs stopped stressed rats from pressing the lever more often for cocaine. Together, these findings suggested that activating κORs in the VTA contributes to stress-induced drug relapse.

Polter et al. have now discovered how stress activates κORs. It turns out that stressful or unpleasant experiences cause the brain to produce a protein called dynorphin, which binds to and activates the κORs. After a stressful event, the receptors are said to have become constitutively active, and blocking this constitutive activity prevents stress from inducing drug-seeking behavior. Polter et al. show that binding of dynorphin is needed to change the shape of the receptors so that they remain active even after dynorphin has detached, but it is likely that other molecules are also involved.

This is the first study to show a link between stress, constitutive activation of κORs, and drug relapse. The next step is to work out why this process occurs on only some and not all occasions when the brain releases dynorphin, and why only certain κORs in the VTA respond in this way. Whether constitutive kOR activity drives stress-related craving in people with a history of drug abuse and how to halt these cravings also remain to be determined.

excitatory synaptic transmission onto both dopaminergic and non-dopaminergic VTA neurons (*Margolis et al., 2005*), reduce inhibitory synaptic transmission in a subset of dopamine neurons (*Ford et al., 2006*) and inhibit somatodendritic dopaminergic IPSCs (*Ford et al., 2007*). VTA κORs also can control the interactions between stress and reward. Our previous work identified a form of stress-sensitive synaptic plasticity at inhibitory synapses on VTA dopamine neurons ($LTP_{GABA}$; *Nugent et al., 2007*, *2009*; *Niehaus et al., 2010*). $LTP_{GABA}$ is induced via activation of nitric oxide synthase in the dopamine neuron, leading to nitric oxide (NO) release, and enhancement of GABA release through cGMP signaling (*Nugent et al., 2007*, *2009*).

More recently, we showed that acute stress blocks $LTP_{GABA}$ through activation of κORs, and that preventing this activation via intra-VTA administration of the κOR antagonist, nor-binaltorphimine (norBNI), prevents stress-induced reinstatement of cocaine-seeking (*Graziane et al., 2013*). Remarkably, a single exposure to stress leads to a loss of $LTP_{GABA}$ that lasts for at least five days and is mediated by persistent activation of VTA κORs (*Polter et al., 2014*). We have also shown that treatment with the κOR antagonist *after* stress can rescue stress-induced reinstatement. These studies highlight the importance of κOR-mediated regulation of LTP at GABAergic synapses in stress-induced drug seeking and underscore the need to better understand the mechanism of this unique and persistent regulation.

In the present study, we have now identified the mechanism by which activation of κORs and suppression of $LTP_{GABA}$ in the VTA is maintained for multiple days after an acute, severe stressor. We present evidence that stress blocks $LTP_{GABA}$ by inducing constitutive activation of κORs at VTA inhibitory synapses rather than through persistent increases in dynorphin release. This constitutive activity is likely to be triggered initially by signaling through the endogenous ligand dynorphin, but then is persistently maintained independently of dynorphin release. In parallel, we find that the persistent drug-seeking induced by a single exposure to acute stress is also dependent on constitutive activity of κORs. Our results reveal a novel mechanism of experience-dependent regulation of κOR

function, and emphasize the essential role of κORs in mediating stress-induced changes in synaptic plasticity and drug-seeking behavior.

## Results

### JNK-dependent rescue of LTP$_{GABA}$ by acute norBNI

As previously shown, bath application of the nitric oxide donor SNAP potentiates GABAergic synapses on dopamine neurons in the VTA, similarly to high-frequency stimulation of VTA afferents; this potentiation is blocked by single exposure to multiple drugs of abuse or acute cold-water swim stress (LTP$_{GABA}$; *Nugent et al., 2007*; *Niehaus et al., 2010*; *Graziane et al., 2013*; *Polter et al., 2014*; *Figure 1A–B*). Our recent studies indicate that blocking κORs with norNBI prevents and reverses the effects of acute stress on LTP$_{GABA}$, even when administered several days after stress (*Graziane et al., 2013*; *Polter et al., 2014*). We therefore investigated whether stress-induced, persistent activation of κORs could be detected ex vivo in the midbrain slice. We subjected rats to acute cold water forced swim stress and prepared midbrain slices 24 hr later. If after stress, κORs in the VTA are persistently signaling in vitro, we reasoned that bath-applied norBNI could be used to rescue SNAP-induced LTP$_{GABA}$. Bath application of norBNI (100 nM) indeed allowed us to elicit NO-dependent LTP$_{GABA}$ in slices from stressed animals (*Figure 1E*), indicating that stress-induced activity of κORs in the VTA persists through brain slice preparation and recovery. It seemed unlikely that sufficient endogenous dynorphin could be released tonically from the denervated brain slices to maintain a block of LTP$_{GABA}$ in vitro. We therefore next sought to establish the mechanism by which norBNI rescued this plasticity. In addition to competing with agonists at the κOR agonist binding site, norBNI acts as an inverse or collateral agonist, and its interactions with the κOR can non-competitively inhibit further activity of κORs via activation of the JNK signaling cascade (*Bruchas et al., 2007*; *Melief et al., 2010*, *2011*). We hypothesized that the rescue of LTP$_{GABA}$ by norBNI might also occur non-competitively via JNK signaling (*Figure 1C*). Slices were treated with the JNK inhibitor SP600125 (20 µM) for 10 min prior to bath application of norBNI (*Figure 1D*). In contrast to the robust SNAP-induced potentiation observed in slices treated with norBNI alone, we found that LTP$_{GABA}$ remained blocked in slices pretreated with SP600125 (*Figure 1F–H*). Importantly, bath application of SP600125 did not interfere with expression of LTP$_{GABA}$ in slices from naïve animals or the loss of LTP$_{GABA}$ in slices from stressed animals (*Figure 1—figure supplement 1A–B*). Therefore, JNK activity has no role in LTP$_{GABA}$ induction or in the block of this plasticity by κORs, but is required for norBNI to rescue LTP$_{GABA}$ following stress.

### LTP$_{GABA}$ is not rescued by a neutral antagonist

Our data indicate that following stress, κOR activation persists even in the brain slice, and is rescued in a JNK-dependent manner. This suggests that non-competitive actions of norBNI, rather than its block of dynorphin binding, are relevant to the loss of LTP$_{GABA}$. To test this hypothesis further, we again utilized pharmacological tools in slices from stressed animals. We treated such slices with either norBNI or 6β-naltrexol, a neutral antagonist that only inhibits agonist-stimulated κOR activity (*Figure 2A–B*; *Wang et al., 2007*). If norBNI rescues LTP$_{GABA}$ only because it can activate JNK signaling, we would predict that a neutral antagonist that only inhibits κOR agonist binding would be ineffective (*Figure 2A*, *Wang et al., 2007*). While norBNI treatment rescued LTP$_{GABA}$, bath application of the neutral antagonist did not reverse the stress-induced block of LTP$_{GABA}$ (*Figure 2C–F*). Bath application of 6β-naltrexol was sufficient to prevent depression of EPSCs onto VTA dopamine neurons induced by the κOR agonist U50488 (*Figure 2—figure supplement 1B*, *Margolis et al., 2005*), indicating that this concentration of the drug is sufficient to block κORs in the VTA slice. 6β-naltrexol did not have any effects on basal inhibitory transmission in slices from stressed or naïve rats (*Figure 2—figure supplement 1A*). These results show that a κOR competitive antagonist cannot effectively rescue LTP$_{GABA}$ following stress, and suggest that the persistent block of LTP$_{GABA}$ is maintained by constitutive activation of κORs in the VTA rather than a prolonged increase in dynorphin release.

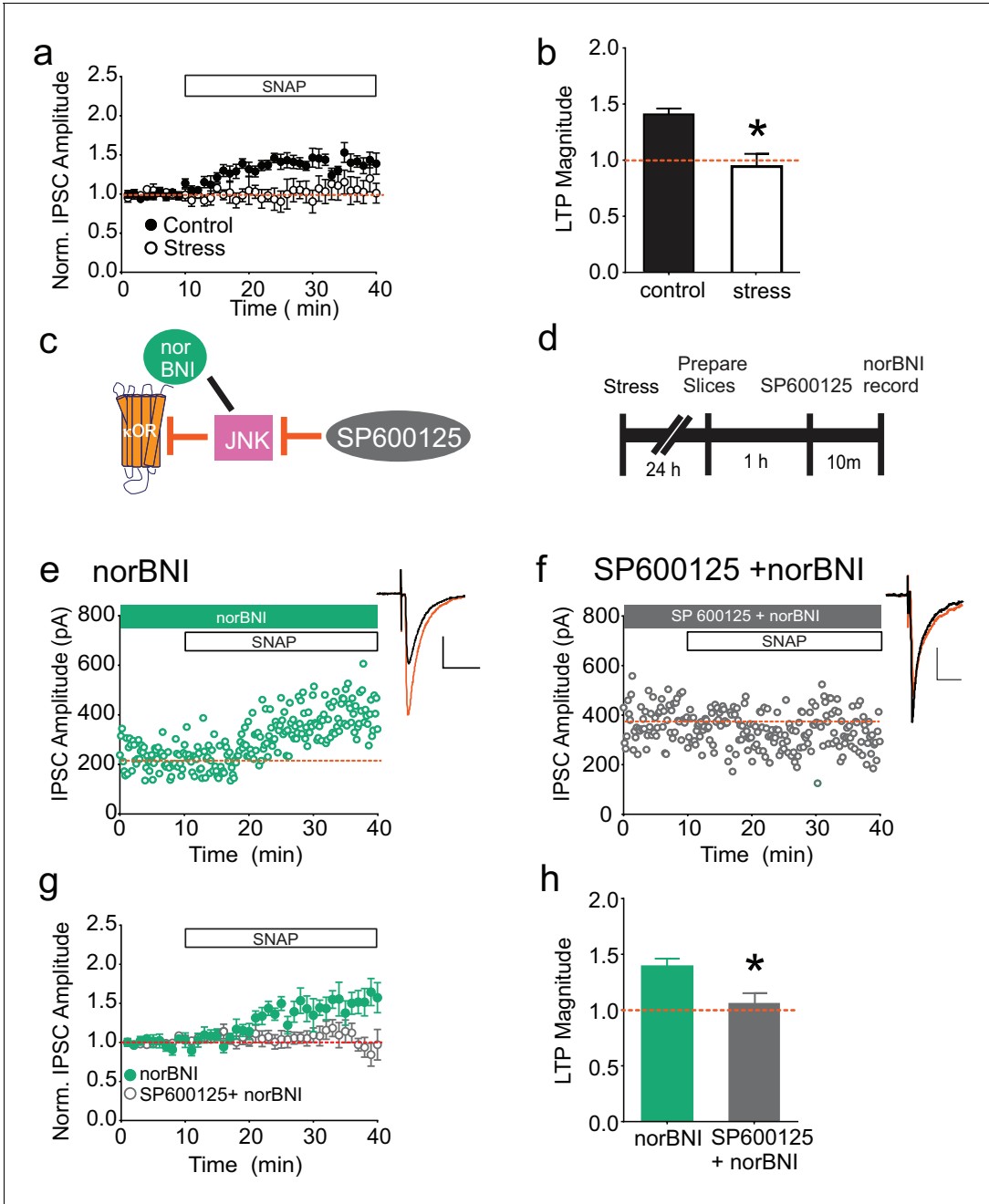

**Figure 1.** norBNI rescues LTP$_{GABA}$ through activation of JNK. (A) Summary data showing the blockade of LTP$_{GABA}$ after stress. (B) Comparison of the magnitude of LTP$_{GABA}$10–15 min after SNAP application. (IPSC amplitudes, control: 140 ± 5% of baseline values, n = 13; stress: 94 ± 11% of baseline values, n = 6; unpaired t-test, *p=0.0005. (C) Schematic of norBNI's competitive and non-competitive inhibition of κOR signaling. (D) Experimental design. (E) Representative single experiment showing that bath application of norBNI (100 nM) rescues LTP$_{GABA}$ in a slice prepared 24 hr after stress. (F) Representative single experiment from a slice prepared 24 hr after stress showing that norBNI does not rescue LTP$_{GABA}$ in the presence of the JNK inhibitor SP600125 (20 μM). (G) Summary data from both groups. (H) Comparison of the magnitude of LTP$_{GABA}$10–15 min after SNAP application. (IPSC amplitudes, norBNI only: 139 ± 7% of baseline values, n = 6; norBNI+SP600125: 106 ± 9% of baseline values, n = 11; unpaired t-test, *p=0.029.) Insets for this and all figures: IPSCs before (black trace, control) and 15 min after drug application (red trace, SNAP, 400 μM). Scale bars: 20 ms, 100 pA. Insets are averages of 12 IPSCs.

The following figure supplement is available for figure 1:

**Figure supplement 1.** Inhibition of JNK does not affect LTP$_{GABA}$ or its block by stress in the absence of norBNI.

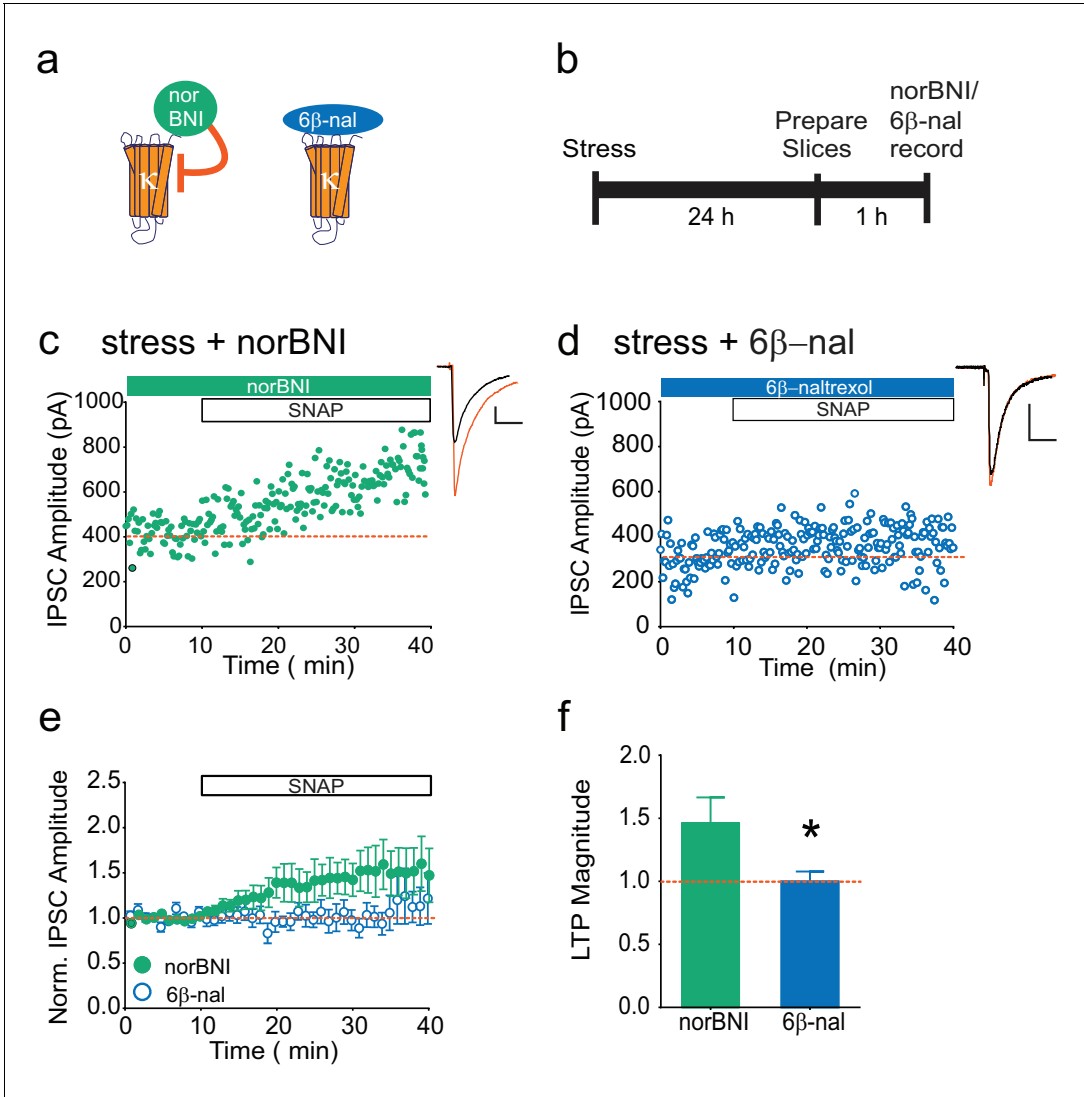

**Figure 2.** The neutral antagonist *6β*-naltrexol fails to rescue LTP$_{GABA}$ in slices from stressed animals. (A) Schematic of norBNI and *6β*-naltrexol inhibition of κOR signaling. (B) Experimental design. (C) Representative experiment showing that bath application of norBNI (100 nM) rescues LTP$_{GABA}$ in a slice prepared 24 hr after stress. (D) Representative experiment from a cell 24 hr after stress showing that *6β*-naltrexol (10 μM) fails to rescue LTP$_{GABA}$. (E) Summary data from both groups. (F) Comparison of the magnitude of LTP$_{GABA}$10–15 min after SNAP application. (IPSC amplitudes, norBNI: 141 ± 20% of baseline values, n = 10; *6β*-naltrexol: 100 ± 8% of baseline values, n = 10; unpaired t-test, *p=0.048).

The following figure supplement is available for figure 2:

**Figure supplement 1.** *6β*-naltrexol does not affect basal inhibitory synaptic transmission but does block κORs.

## Transient κOR activation leads to persistent κOR activity

How might acute stress cause constitutive activation of κORs? While the results of our slice experiments rule out a requirement for elevated dynorphin in maintaining persistent activity of VTA κORs following stress, dynorphin release during or immediately following stress may be needed to trigger a change in the receptor leading to prolonged constitutive activation. If this model is correct, preventing binding of dynorphin to the κOR *during* stress would prevent the loss of LTP$_{GABA}$. However *after* stress, when the block of LTP$_{GABA}$ is no longer dynorphin-dependent, preventing dynorphin binding would not rescue LTP$_{GABA}$. To test this idea, we treated animals with the competitive antagonist *6β*–naltrexol either 30 min before or one day after FSS (*Figure 3A*). Consistent with our hypothesis, cells from animals treated with *6β*-naltrexol before stress exhibited LTP$_{GABA}$, while those

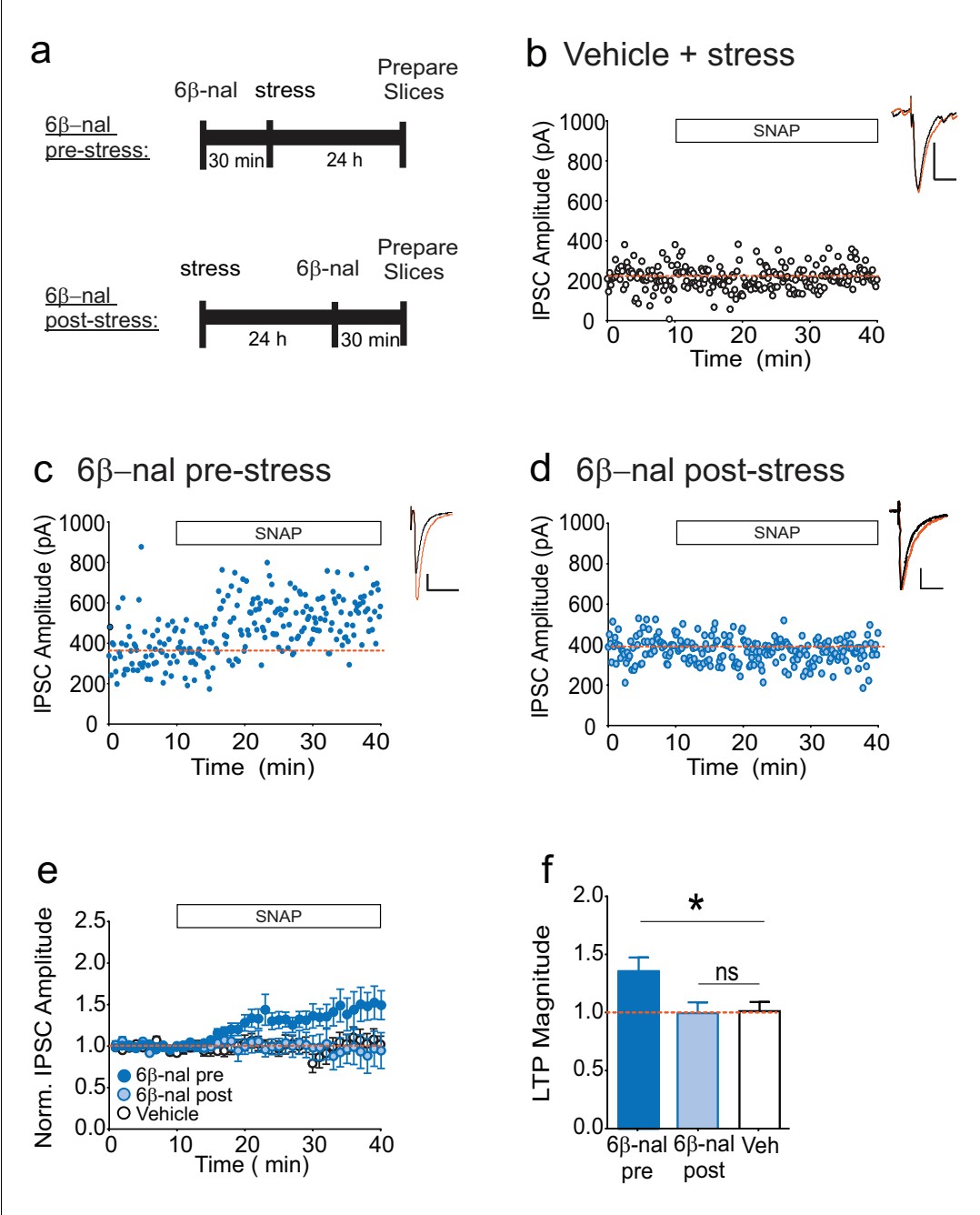

**Figure 3.** 6*β*-naltrexol rescues LTP_GABA when administered pre-stress, but not post-stress. (A) Experimental design. (B) Representative experiment showing that a cell from a vehicle-treated stressed animal does not exhibit LTP_GABA. (C) Representative experiment showing that a cell from an animal treated with 6*β*-naltrexol (10 mg/kg) 30 min pre-stress exhibits LTP_GABA. (D) Representative experiment showing that a cell from an animal treated with 6*β*-naltrexol 24 hr post-stress does not exhibit LTP_GABA. (E) Summary data showing compiled data from all groups. (F) Comparison of the magnitude of LTP_GABA 10–15 min after SNAP application. (1-way ANOVA followed by Dunnett's multiple comparison test. $F_{2, 30}$=4.231,p=0.024. IPSC amplitudes, 6*β*-naltrexol pre-stress: 136 ± 12% of baseline values, n = 12, p<0.05 from vehicle; 6*β*-naltrexol post-stress: 100 ± 9% of baseline values, n = 11, n.s. from vehicle; vehicle+stress: 102 ± 8% of baseline values, n = 10).

treated one day after stress did not, similarly to the vehicle-treated animals (*Figure 3B-F*). In contrast, our previous studies have shown that treating rats with norBNI at the same time point after stress (one day) rescues LTP_GABA (*Polter et al., 2014*). These data strongly support the idea that the persistent block of LTP_GABA following acute swim stress is mediated by dynorphin-dependent

activation of the κOR followed by a transition to dynorphin-independent constitutive activity of the receptor.

To investigate whether a brief activation of κORs is sufficient to produce persistently activated κORs, we treated rats with a single dose of the κOR agonist, U50488, and measured LTP$_{GABA}$ at various time points after injection (*Figure 4A*). Upon injection, U50488 rapidly enters the CNS and is metabolized and undetectable in the brain by 24 hr after administration (*Russell et al., 2014*), and

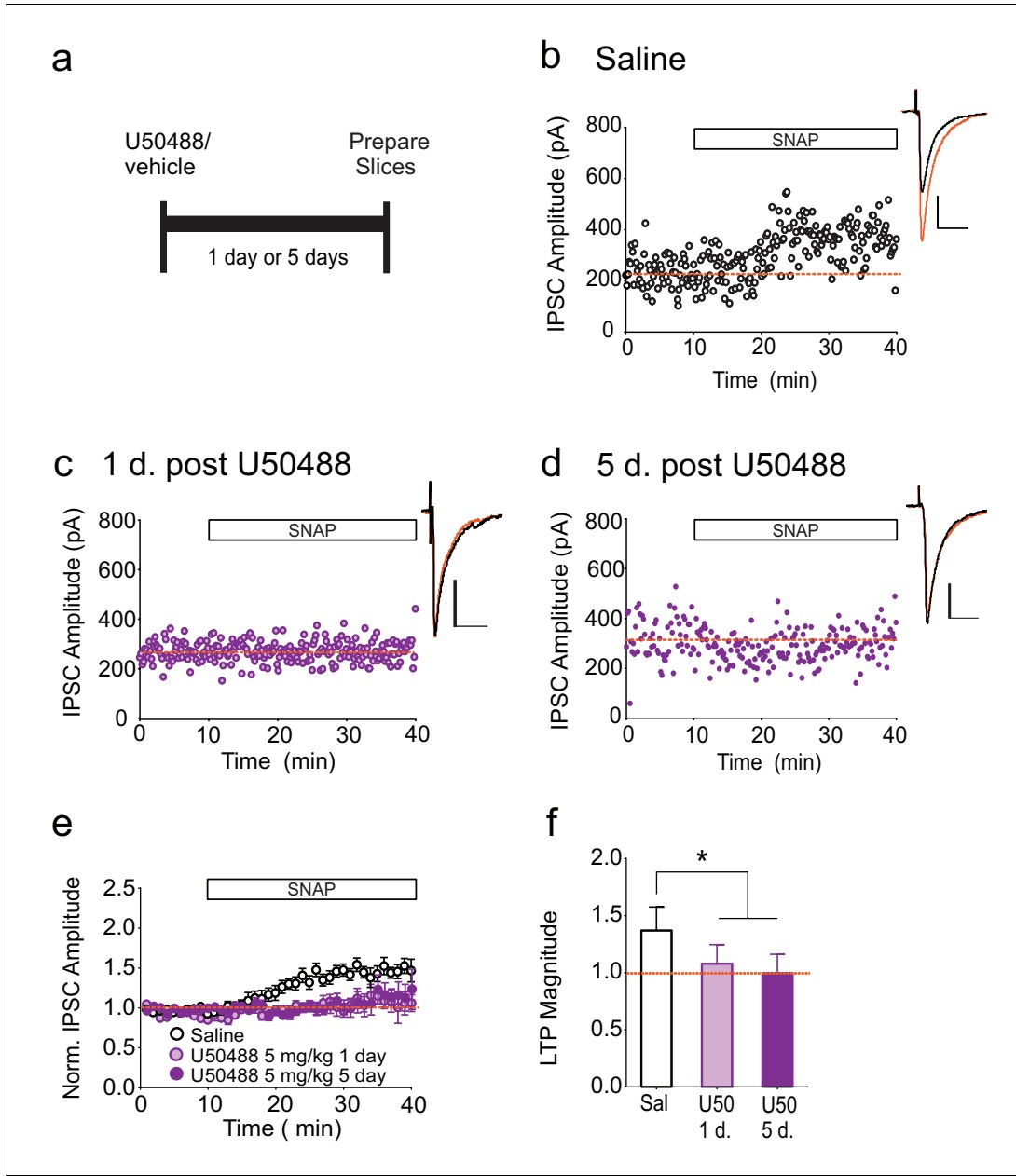

**Figure 4.** Single treatment with a κOR agonist leads to prolonged blockade of LTP$_{GABA}$. (A) Experimental design. (B) Representative experiment showing that a cell from a saline-treated animal exhibits LTP$_{GABA}$. (C) Representative single experiment showing a cell prepared 24 hr after a single treatment with U50488 (5 mg/kg) does not exhibit LTP$_{GABA}$. (D) Representative experiment showing that a cell prepared five days after a single treatment with U50488 does not exhibit LTP$_{GABA}$. (E) Summary data from all groups. (F) Comparison of the magnitude of LTP$_{GABA}$10–15 min after SNAP application. (1-way ANOVA followed by Dunnett's multiple comparison test. F$_{2, 27}$=12.21, p=0.0002. IPSC amplitudes, Saline: 137 ± 6% of baseline values, n = 11; U50488 1 day: 108 ± 6% of baseline values, n = 9, p<0.05 vs. saline; U50488 5 days: 100 ± 5% of baseline values, n = 10, p<0.05 vs. saline).

we therefore expect that within our experimental time frame, U50488 was no longer occupying the κOR. In neurons from saline-treated animals, bath application of SNAP robustly potentiated IPSCs (*Figure 4B*). In contrast, SNAP was unable to elicit LTP$_{GABA}$ in neurons from rats either one or five days after U50488 administration (*Figure 4C–F*). Notably, this time course closely mirrors that of the in vivo block of LTP$_{GABA}$ following acute stress (*Polter et al., 2014*).

## Specificity to inhibitory VTA synapses

We next addressed the question of whether κORs at other brain synapses are also persistently activated after acute stress. Bath application of the κOR agonist U69593 has been reported to depress the amplitude of glutamatergic EPSCs in both VTA I$_h$ positive (presumptive dopamine neurons) and I$_h$ negative (presumptive non-dopamine neurons), and norBNI reverses this depression (*Margolis et al., 2005*). Therefore if κORs at excitatory synapses become constitutively activated after swim stress, reducing their activity with norBNI should be detectable as potentiation of excitatory VTA synapses. To test this, we prepared VTA slices 24 hr after FSS. We recorded EPSCs from I$_h$ positive and I$_h$ negative neurons from both stressed and unstressed animals and bath-applied norBNI. NorBNI had no effect on EPSC amplitude in I$_h$-positive neurons in slices from either naïve or stressed animals (*Figure 5A–C*), and norBNI did not increase EPSC amplitudes in VTA I$_h$-negative neurons in slices from either naïve or stressed animals (*Figure 5D–F*). Therefore, the persistent constitutive κOR activation we observe at GABAergic synapses after acute stress does not occur at all κORs, even within the VTA.

## Constitutive activity of κORs is required for prolonged stress-induced cocaine seeking

Numerous studies from our lab and others have shown a close association between κOR activation and stress-induced drug-seeking behavior (*McLaughlin et al., 2003*; *Redila and Chavkin, 2008*; *Land et al., 2009*; *Wee and Koob, 2010*; *Graziane et al., 2013*; *Zhou et al., 2013*; *Polter et al., 2014*). We recently reported that blocking κORs with norBNI reverses the modest but prolonged reinstatement of cocaine-seeking induced by swim stress (*Conrad et al., 2010*; *Graziane et al., 2013*). This rescue is seen even when norBNI is administered two hours after stress (*Polter et al., 2014*). These findings are consistent with the hypothesis that reinstatement of cocaine-seeking after swim stress requires activation of VTA κORs and suppression of LTP$_{GABA}$. Having now shown that the blockade of LTP$_{GABA}$ by swim stress is dependent on constitutive activity of κORs, we next tested whether reinstatement of cocaine seeking is similarly dependent on constitutively active κORs.

Rats were trained to self-administer cocaine for a minimum of 10 days. Animals then underwent extinction training, and after the final extinction session, they were subjected to forced swim stress, and then returned to their home cages. Twenty-four hours after stress, one group of animals was treated with norBNI and a second group with saline. A third group was treated with 6β–naltrexol 2 days after stress and 60 min prior to reinstatement testing (*Figure 6A*). Due to the differing pharmacokinetic profiles of norBNI and 6β–naltrexol, time of administration was varied to optimize block of the κOR during the reinstatement test and to ensure that all animals were tested for reinstatement at the same time point (*Endoh et al., 1992*; *Raehal et al., 2005*); thus, all animals were tested for reinstatement 48 hr after stress.

As previously shown, after acute stress, vehicle-treated animals showed a significant elevation of lever pressing compared to the final extinction session (*Figure 6B*). Although the reinstatement was modest, this was measured two full days after the stress, demonstrating the prolonged increase in cocaine-seeking (*Conrad et al., 2010*). In contrast, animals given norBNI 24 hr post-stress did not increase their lever pressing two days after stress (*Figure 6B*). Moreover, the neutral antagonist 6β-naltrexol did not prevent reinstatement, as 6β–naltrexol treated animals significantly increased lever pressing compared to the final extinction session (*Figure 6B*). These data suggest that while persistent activation of κORs underlies the prolonged reinstatement induced by swim stress, this is mediated by constitutively active receptors rather than by long-term increases in the level of the endogenous ligand dynorphin.

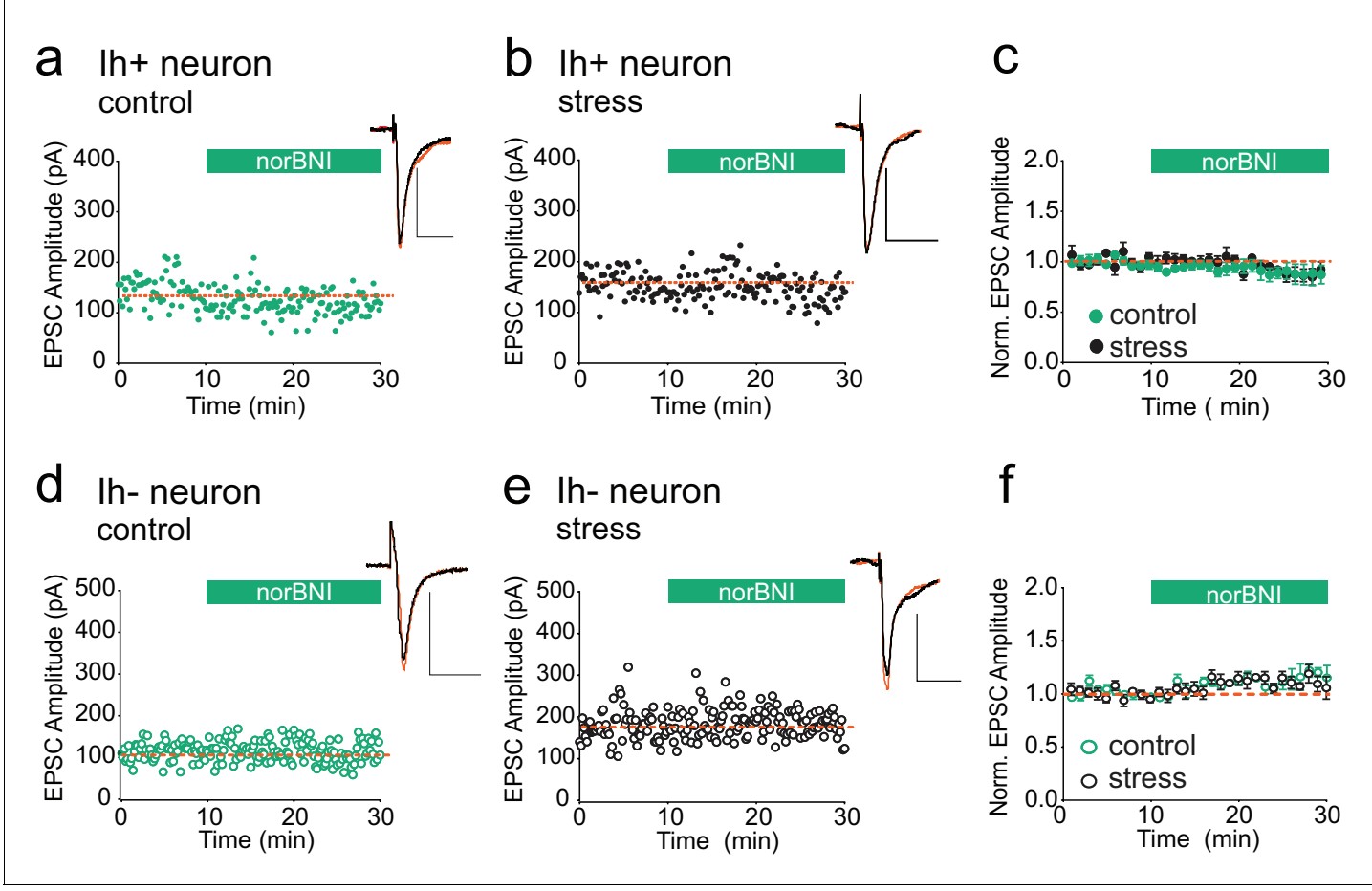

**Figure 5.** κORs at VTA excitatory synapses are not constitutively activated by stress. (**A**) Representative experiment showing that norBNI (100 nM) does not potentiate excitatory synapses on $I_h$+ VTA neurons in a slice prepared from a control animal. (**B**) Representative experiment showing that norBNI does not potentiate excitatory synapses on $I_h$+ VTA neurons in a slice prepared from a stressed animal. (**C**) Summary data from $I_h$+ neurons. No significant difference in IPSC amplitude 10–15 min after norBNI application (t-test p=0.81 IPSC amplitudes, control: 94 ± 2% of baseline values, n = 5; stressed: 92 ± 6% of baseline values, n = 6). (**D**) Representative experiment showing that norBNI does not potentiate excitatory synapses on $I_h$− VTA neurons in a slice prepared from a control animal. (**E**) Representative single experiment showing that norBNI does not potentiate excitatory synapses on $I_h$− VTA neurons in a slice prepared from a stressed animals. (**F**) Summary data from $I_h$− neurons. No significant difference in IPSC amplitude 10–15 min after norBNI application (t-test p=0.49 IPSC amplitudes, control: 112 ± 2% of baseline values, n = 5; stressed: 110 ± 4% of baseline values, n = 5).

## Discussion

Acute stress causes a loss of plasticity at VTA GABA$_A$ synapses that lasts for days and is caused by persistent activation of κORs (*Graziane et al., 2013*; *Polter et al., 2014*). This activation could be caused either by a prolonged increase in dynorphin or by an increase in constitutive activity of κORs. In this study, our data support the latter mechanism: a single exposure to an acute stressor causes a lasting constitutive activation of VTA κORs that suppresses plasticity at inhibitory synapses correlated with stress-induced drug-seeking (*Figure 7*). While previous studies have demonstrated constitutive activity of κORs in cultured cells and in the rat brain (*Wang et al., 2007*; *Sirohi and Walker, 2015*), ours is the first demonstration of experience-induced changes in constitutive activity of these receptors. This represents a novel mechanism of regulation by acute stress of the dynorphin-κOR system, and sheds new light on signaling pathways involved in reinstatement of drug seeking.

### Constitutive activation of VTA κORs

It is widely accepted that GPCRs can adopt agonist-independent conformations that are constitutively active (*Seifert and Wenzel-Seifert, 2002*; *Sadée et al., 2005*; *Young et al., 2013 Meye et al.,*

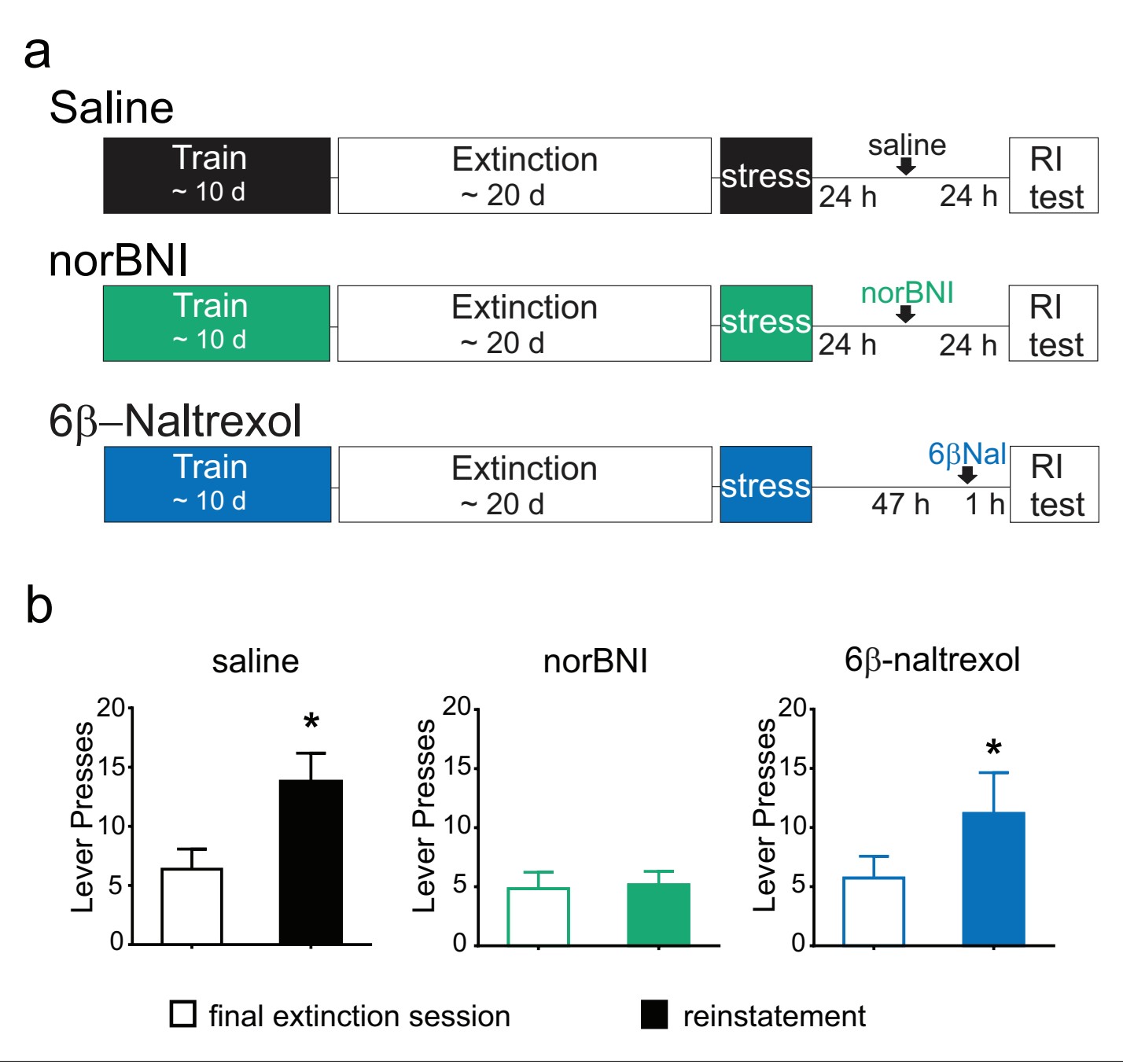

**Figure 6.** Post-stress rescue of reinstatement by norBNI but not *6β*-naltrexol. (**A**) Experimental design. (**B**) Lever pressing during the final extinction session (white bar) and reinstatement session (colored bar). Saline (black): last extinction session: 6.4 ± 1.7 lever presses; reinstatement session: 13.8 ± 2.4 lever presses; n = 8, *p=0.011, paired t-test. norBNI (green): last extinction session: 4.8 ± 1.4 lever presses; reinstatement session: 5.2 ± 1.1 lever presses; n = 6, p=0.76, paired t-test. *6β*-naltrexol (blue): last extinction session: 5.7 ± 1.8 lever presses; reinstatement session: 11.2 ± 3.4 lever presses; n = 11, *p=0.033, paired t-test.

*2014*). In addition to κORs, the other members of the opioid receptor subfamily, μOR and δOR, have both been shown to exhibit constitutive activity (*Wang et al., 1994*, *2004*; *Chiu et al., 1996*; *Wang et al., 1999*; *Liu and Prather, 2001*; *Wang et al., 2007*; *Corder et al., 2013*). κORs themselves have been shown to exhibit constitutive activity, both in heterologous expression systems (*Becker et al., 1999*; *Wang et al., 2007*) and in the rat PFC (*Sirohi and Walker, 2015*). A decrease

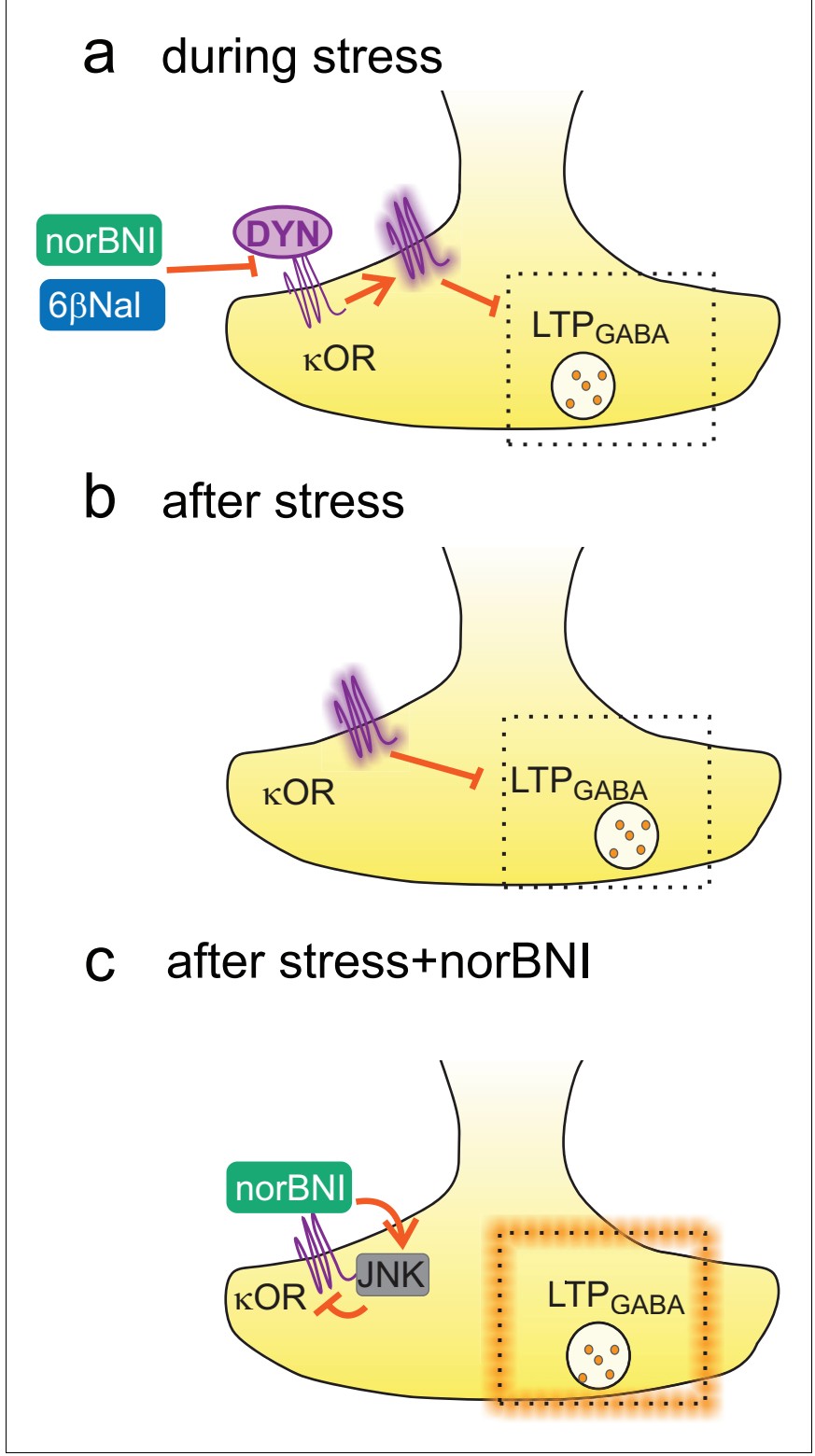

**Figure 7.** Constitutive activation of κORs by stress. (**A**) During stress, dynorphin binding to the κOR triggers a shift to a constitutively active state. By blocking dynorphin binding, both norBNI and 6β-naltrexol prevent the loss of LTP$_{GABA}$ during this time. (**B**) After stress, the block of LTP$_{GABA}$ is maintained by constitutive activity of κORs and is no longer dependent on dynorphin binding. (**C**) norBNI reverses the stress-induced block of LTP$_{GABA}$ by activating the JNK signaling pathway which non-competitively reduces κOR activity.

in fear and anxiety behaviors in rats after acute footshock has also been reported that is reduced by post-shock norBNI, suggesting the possibility of constitutive κOR activation, although a high dose was required and the effect of norBNI was not compared to a neutral antagonist (*Rogala et al., 2012*). Very little is known, however, about the processes that regulate transitions between constitutively active and inactive states, presumably representing distinct receptor conformations (*Seifert and Wenzel-Seifert, 2002*; *Sadée et al., 2005*).

We present two critical pieces of data indicating that stress induces constitutive activity of κORs. First, brief application (~15 min) of a κOR inverse agonist to VTA slices from stressed rats rescues LTP$_{GABA}$ in a JNK-dependent fashion. Second, the neutral antagonist does not rescue LTP$_{GABA}$. Signaling through the JNK pathway is thought to be responsible for the long-lasting non-competitive inhibition of the κOR (*Bruchas et al., 2007*; *Melief et al., 2010*, *2011*). In our experiments, the requirement of JNK for norBNI to rescue LTP$_{GABA}$ is evidence that norBNI acts through non-competitive means, and suggests that the persistent activation of the κOR following stress does not require continuous receptor binding by ligand. Importantly, inhibition of JNK signaling alone did not prevent LTP$_{GABA}$ induction in slices from naïve animals, nor did it restore LTP$_{GABA}$ in slices from stressed animals, indicating that the role of JNK is limited to inhibition of the receptor by norBNI. The failure of the JNK inhibitor to rescue LTP$_{GABA}$ indicates that inhibition of LTP$_{GABA}$ by κORs is not mediated by the JNK pathway, but instead most likely through one of the other pathways downstream of κORs, such as the p38 or ERK MAPK pathways, or through activation of Gα$_i$ (*Bruchas and Chavkin, 2010*; *Iñiguez et al., 2010*; *Ehrich et al., 2015*).

The inability of the neutral antagonist, 6β–naltrexol, to rescue LTP$_{GABA}$ is consistent with stress promoting constitutive κOR activity at inhibitory VTA synapses. NorBNI, through activation of JNK, reduces the signaling capacity of the κOR regardless of whether this occurs through constitutive activity or increased dynorphin binding. In contrast, a neutral antagonist like 6β-naltrexol could only reverse the loss of LTP$_{GABA}$ by preventing agonist binding to the receptor. In contrast to the rapid restoration of LTP$_{GABA}$ by bath application of norBNI, bath application of 6β–naltrexol did not rescue LTP$_{GABA}$. This discrepancy cannot be explained by insufficient concentration or time of application of 6β–naltrexol, as a similar bath perfusion of 6β–naltrexol was sufficient to block the depression of EPSCs onto VTA dopamine neurons induced by bath application of the κOR agonist, U50488. The simplest explanation of our data is that acute stress induces constitutive activity of the κOR. Alternatively, norBNI may promote JNK signaling via an unknown mechanism independent of κOR receptors.

## Experience-induced constitutive activity

Acute stress appears to trigger a shift towards constitutively active κORs through a transient release of the endogenous κOR ligand, dynorphin (*Figure 7*). Our strongest evidence for this model is the ability of the neutral antagonist 6β–naltrexol to prevent the loss of LTP$_{GABA}$ when administered before, but not after stress. Although 6β–naltrexol has equivalent affinity for μ and κ ORs, (*Wang et al., 2007*) our previous work has shown that the block of LTP$_{GABA}$ by stress is unaffected by pre-stress administration of the μOR antagonist cyprodime (*Graziane et al., 2013*). Therefore, the ability of 6β–naltrexol to prevent the stress-induced loss of LTP$_{GABA}$ is unlikely to involve μOR signaling and instead occurs by blocking dynorphin binding to the κOR. A single in vivo systemic administration of the κOR agonist U50488 also blocks LTP$_{GABA}$ for at least five days, supporting the idea that brief agonist exposure alone is sufficient to trigger lasting constitutive κOR activity.

How might activation of κORs by its endogenous ligand shift the receptor towards constitutive activity? In a heterologous cell-culture system, previous exposure to a κOR agonist alone significantly increased constitutive activity of the receptor (*Wang et al., 2007*). More is known regarding regulation of constitutive activity of μORs. In either cultured cells heterologously expressing μORs (*Wang et al., 1994*, *2000*; *Liu and Prather, 2001*) or in intact animals (*Wang et al., 2004*; *Shoblock and Maidment, 2006*; *Meye et al., 2012*), exposure to the μOR agonist morphine triggers an increase in constitutive activity of μORs. Morphine-induced constitutive activity of μORs is regulated by calmodulin and protein kinases. Under basal conditions, calmodulin binding to the μORs prevents constitutive association with G-proteins. Following morphine exposure, calmodulin dissociates from the μOR, allowing constitutive activation (*Wang et al., 1999*, *2000*). Although it is unknown whether calmodulin regulates the activity of κORs, an intricate scaffolding complex regulates κOR signaling (*Bruchas and Chavkin, 2010*), and future studies investigating the role of these

signaling complexes in κOR activity in response to stress will be important and intriguing. Most of the work investigating constitutive activation of GPCRs has focused on enhancement of constitutive activity by administration of exogenous ligands (*Meye et al., 2014*), while considerably less is known about induction of constitutively active states of GPCRs by endogenous signaling. However, it was recently reported that inflammatory pain increases constitutively active μORs in the mouse spinal cord, leading to hyperalgesia and cellular dependence (*Corder et al., 2013*).

Our data indicate that treatment with the κOR agonist U50488 alone is sufficient to produce constitutively active κORs on inhibitory VTA synapses. A remaining puzzle is why κOR activation by endogenous ligand can produce constitutively active receptors at some synapses but not at their neighbors, and in response to certain environmental cues (acute stress) but not to others during which dynorphin may also be released. One possibility is that receptors in different cell types may couple to different signaling cascades or scaffolding molecules. Another possibility is that coordinated signaling between κORs and another neurotransmitter system may be required. Our prior work indicates that activation of glucocorticoid receptors is required for the block of LTP$_{GABA}$ by stress (*Niehaus et al., 2010*; *Polter et al., 2014*). Although persistent activation of these receptors is not seen after stress, it is possible that coincident activation of glucocorticoid and kappa opioid receptors leads to constitutive activation of the latter. Additionally, it has been reported that the orexin-1 receptor attenuates κOR inhibition of cAMP production, but enhances recruitment of $\beta$-arrestin and p38 MAPK activation, and both effects are prevented by the JNK inhibitor SP600125 used in our study (*Robinson and McDonald, 2015*). Both orexin and dynorphin are co-released from hypothalamic projections to the VTA (*Chou et al., 2001*; *Muschamp et al., 2014*; *Baimel et al., 2015*). This arrangement raises the possibility that release of both peptides together, or perhaps simultaneous release of dynorphin and an unknown neurotransmitter acting similarly to orexin, may initiate signaling events not triggered by dynorphin alone. The putative dual receptor signaling might be one way to induce synapse- or experience- selective constitutive κOR activity.

## Regulation of LTP$_{GABA}$ by κORs

One unanswered question is how κORs suppress the expression of LTP$_{GABA}$. Our previous studies have shown that LTP$_{GABA}$ is triggered by nitric oxide-mediated activation of cGMP-protein kinase G (PKG) signaling (*Nugent et al., 2007*, *2009*; *Niehaus et al., 2010*). Because an exogenous source of nitric oxide (SNAP) does not rescue potentiation following stress, the blockade is likely to occur in the presynaptic terminal between activation of guanylate cyclase and enhancement of GABAergic release. While it is possible that κOR activation generally depresses GABA release, our previous work (*Graziane et al., 2013*) found no change in mIPSC frequency following cold water swim stress. These data suggest that κORs do not alter basal GABA release. Moreover, LTP$_{GABA}$ is also lost 24 hr after a single morphine exposure, and in this situation a cGMP analog or strong activation of sGC potentiates GABA release (*Nugent et al., 2007*; *Niehaus et al., 2010*). We therefore favor a mechanism by which after acute stress, constitutively-active κORs similarly act on a substrate that limits induction of plasticity without affecting basal release mechanisms, perhaps through downregulation of soluble guanylyl cyclase, or scaffolding changes that sequester PKG from its substrates.

Although it remains unknown under what conditions LTP$_{GABA}$ is activated in an intact animal, our prior studies shed some light on its potential roles. LTP$_{GABA}$ is a heterosynaptic form of plasticity that can be triggered by a high-frequency tetanus that activates NMDAR-dependent activation of calcium-sensitive nitric oxide synthase (*Nugent et al., 2007*). We therefore expect that LTP$_{GABA}$ would be induced when there is robust activation of excitatory inputs onto dopamine neurons. LTP$_{GABA}$ may play a homeostatic role to enhance inhibition of dopamine neurons after strong NMDAR-activating excitation. Loss of LTP$_{GABA}$, therefore, would result in an imbalance between inhibitory and excitatory input onto the dopamine neuron. As GABAergic synapses on dopamine neurons strongly control their spontaneous firing (*van Zessen et al., 2012*), the loss of LTP$_{GABA}$ is likely to prolong or enhance firing in response to salient stimuli.

## Constitutive activity of κORs in the VTA and drug-seeking behavior

The critical role of the VTA in reinstatement of drug seeking has been repeatedly underscored (*McFarland et al., 2004*; *Briand et al., 2010*; *Graziane et al., 2013*; *Mantsch et al., 2016*), and

within the VTA, GABAergic synapses on VTA dopamine neurons powerfully regulate DA cell firing (*Tan et al., 2012*; *van Zessen et al., 2012*; *Polter and Kauer, 2014*).

Our work shows that stress produces long-lasting κOR constitutive activity that is restricted to inhibitory synapses on dopamine cells, thereby affecting information stored or processed here for far longer than at excitatory synapses. We might therefore predict two phases of stress-induced κOR activation. We hypothesize that dynorphin is released during and/or immediately after stress, depressing EPSCs onto dopaminergic neurons and hyperpolarizing dopaminergic neurons, on balance decreasing dopaminergic neuron excitability (*Margolis et al., 2003*, *2005*; *Ford et al., 2006*). However, as dynorphin is degraded, we would expect that the strength of excitatory synapses would return to normal levels while LTP$_{GABA}$ would become blocked by constitutive activity of κORs, a state lasting at least five days after swim stress. This would instead increase the firing rate of dopaminergic neurons, particularly in response to excitatory stimuli. This increased excitability could contribute to the increased drive towards drug-seeking behavior upon exposure to spatial cues associated with past drug experience (i.e., return to the operant chamber), and could create a state of vulnerability to further stressors. Indeed, in rats subjected to the same cold water stress used in this study, the firing rate of dopaminergic neurons remains elevated for several days afterwards (*Marinelli, 2007*). Interestingly, a single dose of the κOR agonist, salvinorin A, has biphasic effects on reward function: immediately after administration, rats exhibit an anhedonic increase in reward thresholds to intracranial self-stimulation. However, 24 hr after salvinorin A administration, rats exhibit *decreased* reward thresholds, indicating an increase in reward sensitivity (*Potter et al., 2011*). This biphasic effect is consistent with a split between short- and long-term effects of κOR activation, perhaps due to differential mechanisms of regulation and constitutive activation of subsets of receptors.

The circuitry of the VTA is highly complex, and dopamine neurons within the VTA exhibit physiological and functional heterogeneity that correlates with projection target. While disagreement remains about the most appropriate pharmacological, physiological, and anatomical markers of different subclasses of dopamine neurons (*Ford et al., 2006*; *Margolis et al., 2006*; *Lammel et al., 2008*, *2011*; *Ungless and Grace, 2012*; *Baimel et al., 2017*), the electrophysiological markers used here and the lateral location of our recordings within the VTA suggest to us that our population of cells may largely comprise dopamine neurons that project to the nucleus accumbens. This may be significant for drug reward, as activation of these neurons has been shown to be rewarding in mice (*Lammel et al., 2012*). Therefore, our study indicates that an acute stressor induces a long-lasting loss of inhibitory plasticity in circuitry that may drive rewarding behavior.

GABAergic afferents on VTA dopamine neurons can release GABA onto either GABA$_A$ or GABA$_B$ receptors. Previous studies including more recent optogenetic approaches have suggested that GABA$_B$ receptor-targeting neurons arise from the nucleus accumbens and regulate drug-induced behaviors (*Sugita et al., 1992*; *Cameron and Williams, 1993*; *McCall et al., 2017*; *Edwards et al., 2017*). However, our earlier work found no LTP$_{GABA}$ at GABA$_B$ synapses on dopamine neurons (*Nugent et al., 2009*), suggesting that the effects of persistently activated κORs are unlikely to involve the nucleus accumbens-VTA GABAergic afferents.

Our data provide the first demonstration that constitutively active κORs in the VTA are required for stress-induced reinstatement of cocaine-seeking. Post-stress (at least 24 hr) administration of norBNI prevents reinstatement, while post-stress administration of the neutral antagonist 6β-naltrexol does not. The ability of norBNI to modify drug-seeking behavior even when given significantly after the stressor is remarkable, and indicates the therapeutic potential of targeting κORs to reverse stress-induced neuroadaptations. The failure of 6β–naltrexol to prevent reinstatement at time points when norBNI is effective strongly suggests that the persistent increase in drug seeking induced by swim stress is mediated by constitutive activity of κORs rather than a prolonged increase in dynorphin release. Furthermore, this result is consistent with an important role for GABAergic synapse plasticity in stress-induced drug-seeking behavior. While considerable attention has been given to the role of LTP at excitatory synapses in the VTA, the κOR block by norBNI does not prevent stress from potentiating excitatory synapses on dopamine neurons (*Graziane et al., 2013*). Our current work confirms that the loss of LTP at GABAergic synapses in the VTA is highly correlated with stress-induced drug-seeking.

## κORs as targets for substance abuse

κORs have shown promise as a potential drug target for substance use and mood disorders (*Bruchas et al., 2010*; *Van't Veer and Carlezon, 2013*; *Crowley and Kash, 2015*) and our work suggests a novel way in which κOR signaling may go awry. In preclinical models, κOR antagonists have shown potential efficacy for depression and for compulsive and stress-induced drug use (*Bruchas et al., 2010*; *Wee and Koob, 2010*). Numerous clinical trials are in progress using κOR ligands to target substance use disorders and depression (*Ehrich et al., 2015*; *Karp et al., 2014*; *Chavkin and Koob, 2016*; *Ling et al., 2016*; *Nasser et al., 2016*). However, many of these trials use buprenorphine, a partial agonist at κORs, or novel compounds which may lack inverse agonist activity, neither of which would reduce activity of constitutive κORs (*Karp et al., 2014*; *Rorick-Kehn et al., 2014*).

Our study suggests that future drug development should consider excess κOR activity through receptor signaling as well as at the level of ligand binding. Similarly, disappointing results or minimal effects in clinical trials may not represent failure of κORs as a pharmaceutical target, but a need to consider drugs that target specific conformations of the κOR that promote constitutive signaling. An alternative strategy would be to target JNK signaling in the VTA, as norBNI appears to rescue κOR function by activating JNK. An intriguing implication of our studies comes from our data that constitutive activity of κORs at inhibitory synapses in the VTA lasts only five to ten days following acute stress (*Polter et al., 2014*), a considerably shorter time period than the 14–21 days typical for turnover of κORs (*McLaughlin et al., 2004*). This suggests that constitutive activity of the κORs may be terminated by an unidentified active mechanism. Future studies investigating such a mechanism could identify targets that could be recruited to promote resilience to stress. Our work demonstrates a novel mechanism of experience-dependent regulation of κORs, and highlights the ability of modulation of κORs to reverse stress-induced neuroadaptations and behavioral deficiencies well after the stressor has occurred. Further study of the mechanisms of constitutive activation of κORs may yield numerous potential targets for the treatment of substance use disorders and other stress-linked illnesses.

# Materials and methods

## Animals

All procedures were carried out in accordance with the guidelines of the National Institutes of Health for animal care and use, and were approved by the Brown University Institutional Animal Care and Use Committee or by the University of Wyoming Institutional Animal Care and Use Committee. For slice electrophysiology studies, male and female Sprague-Dawley rats (P16-25) were maintained on a 12 hr light / dark cycle and provided food and water *ad libitum*. For self-administration studies, male Sprague-Dawley rats (350–450g) were bred in-house and individually housed in a temperature-controlled room with a 12 hr reverse light/dark cycle. All animals were given *ad libitum* access to water throughout experimentation, except during times in which they were in the operant chambers (described below). Rats were 60–70 days old at the start of behavioral experiments.

## Acute forced swim stress

Stress was administered by a modified Porsolt forced swim task (*Niehaus et al., 2010*). Rats were placed for 5 min in cold water (4–6°C), then dried and allowed to recover in a warmed cage for two hours before returning to the home cage. U50488 (5 mg/kg) and 6β-naltrexol (10 mg/kg) were dissolved in saline or 10% DMSO in saline, respectively. Vehicle-injected animals were given an injection of the equivalent volume. For some experiments, animals given vehicle injections at varying time points were collapsed into a single group. Brain slices were prepared at several time points after stress exposure, as described below.

## Preparation of brain slices

Horizontal midbrain slices (250 μm) were prepared as previously described from deeply anesthetized Sprague-Dawley rats (*Nugent et al., 2007*; *Niehaus et al., 2010*; *Polter et al., 2014*). Slices were stored for at least 1 hr at 34°C in oxygenated HEPES holding solution (in mM): 86 NaCl, 2.5 KCl, 1.2 NaH$_2$PO$_4$, 35 NaHCO$_3$, 20 HEPES, 25 glucose, 5 sodium ascorbate, 2 thiourea, 3 sodium pyruvate, 1

MgSO$_4$.7H$_2$O, 2 CaCl$_2$.2H$_2$O (*Ting et al., 2014*). Slices were then transferred to a recording chamber where they were submerged in ACSF containing (in mM): 126 NaCl, 21.4 NaHCO$_3$, 2.5 KCl, 1.2 NaH$_2$PO$_4$, 2.4 CaCl$_2$, 1.0 MgSO$_4$, 11.1 glucose.

## Electrophysiology

General methods were as previously reported (*Niehaus et al., 2010*; *Polter et al., 2014*). Midbrain slices were continuously perfused at 1.5–2 mL / min. Patch pipettes were filled with (in mM): 125 KCl, 2.8 NaCl, 2 MgCl$_2$, 2 ATP-Na+, 0.3 GTP-Na+, 0.6 EGTA, and 10 HEPES. To record IPSCs, the extracellular solution was ACSF (28–32°C) containing: 6,7-dinitroquinoxaline- 2,3-dione (DNQX; 10 µM) and strychnine (1 µM), to block AMPA and glycine receptors respectively. To record EPSCs, 100 µM picrotoxin was added to the ACSF. Dopaminergic neurons, which comprise about 70% of all VTA neurons, were identified by the presence of a large I$_h$ (>50 pA) during a voltage step from −50 mV to −100 mV. GABA$_A$ receptor-mediated IPSCs were stimulated using a bipolar stainless steel stimulating electrode placed 100–300 µm rostral to the recording site in the VTA. Cells were voltage-clamped at −70 mV and input resistance and series resistance were monitored throughout the experiment; cells were discarded if these values changed by more than 15% during the experiment.

## NO-triggered LTP

3-isobutyl-1-methylxanthine (IBMX; 100 µM) was used to inhibit phosphodiesterase-mediated degradation of cGMP and applied via perfused ACSF for at least 10 min prior to induction of LTP$_{GABA}$ by application of the NO donor, SNAP (S-nitroso-N-acetylpenicillamine, 400 µM). Control animals (vehicle-injected stressed or unstressed animals) were interleaved with experimental animals (drug-injected stressed animals). Where indicated, NorBNI (100 nM), 6$\beta$-naltrexol (10 µM), and SP600135 (20 µM) were bath applied to slices at least 10 min prior to induction of LTP$_{GABA}$.

## Self-administration

Rats were anesthetized with ketamine HCl (87 mg/kg, i.m.) and xylazine (13 mg/kg, i.m.) and implanted with intravenous jugular catheters. In order to protect against infection and maintain catheter patency, catheters were flushed daily with 0.2 mL of a mixed cefazolin (0.1 gm/ml) and heparin (100 IU) solution. Rats were allowed to recover for one week before behavioral testing. All self-administration procedures were conducted in standard operant chambers (Med Associates, St. Albans, VT; 30.5 cm x 24.1 cm x 21.0 cm). Each box contained a house light (illuminated throughout behavioral testing) two retractable levers, a cue light, and tone generator. Prior to beginning cocaine self-administration training, animals were food deprived for 24 hr and subsequently placed into the operant chambers overnight for 14 hr. During this session, a response to the active lever (the left lever) resulted in the delivery of a single 45 mg food pellet (#F0165, Bio-Serv, Flemington, NJ) and the presentation of a compound cue (illumination of light above the active lever +5 s tone, 2900 Hz), followed by a 25 s timeout period. Responses to the inactive lever (the right lever) had no programmed consequences but were recorded. Total rewards received were also recorded. On the next day, cocaine self-administration training began. During this time, a response to the active lever yielded a 0.05 ml infusion of 0.20 mg of cocaine (dissolved in 0.9% saline) as well as the presentation of the compound cue (light + tone). Self-administration continued (2 hr/daily) until animals reliably pressed the active lever (3d with minimum of 10 cocaine infusions received). Following the acquisition of cocaine self-administration, all animals underwent extinction training, during which responses to the previously active lever yielded the compound cue but no longer produced drug infusion. Animals were food-restricted to 80% of their body weight during self-administration training. During extinction, animals were given *ad libitum* access to food in the home cage. Extinction procedures continued until animals reached extinction criteria (3d with less than 10 active lever presses).

## Forced swim stress and reinstatement

The day following the last extinction session, rats were subjected to a 5 min forced swim stress in cold water (4-6°C, *Saal et al., 2003*; *Niehaus et al., 2010*). Rats were then split into three groups, receiving (i.p.) injections of either saline (1 ml/kg), norBNI (10 mg/kg), or 6$\beta$-Naltrexol (10 mg/kg). 24 hr after swim stress, rats in the saline and norBNI groups were injected and left undisturbed in their home cage for one day. Rats in the 6$\beta$-naltrexol group were injected one hour prior to reinstatement

testing. At 48 hr after swim stress, all animals were subjected to reinstatement testing, which similarly to extinction yielded the compound cue but no drug infusion.

## Analysis

Magnitude of LTP was determined as mean IPSC amplitude for 5 min just before application of SNAP compared with mean IPSC amplitude from 10–15 min after SNAP application, unless otherwise noted. Data are presented as means ± SEM of the percent IPSC amplitude normalized to IPSCs in the 10 min prior to SNAP application. Statistical methods were not used to determine sample size. Sample size was based on our prior experience and previously published studies (*Graziane et al., 2013*; *Polter et al., 2014*). All reported n's are the number of animals (biological replicates), unless otherwise noted. Significance was determined using a two-tailed Student's *t*-test or a one-way ANOVA with a significance level of $p < 0.05$. All post-hoc comparisons were done using Dunnett's test unless otherwise noted. Self-administration data were analyzed using paired t-tests.

## Materials

IBMX was obtained from Enzo Life Sciences. NorBNI, U50488, and SNAP were obtained from Tocris Biosciences. DNQX, picrotoxin, strychnine, and $6\beta$-naltrexol were obtained from Sigma-Aldrich. SP600125 was obtained from Calbiochem.

## Acknowledgements

The authors would like to thank Dr. Jennifer Whistler for helpful discussion of κOR antagonists, Dr. Michael Bruchas for the suggestion that we test the role of JNK, and Ayumi Tsuda and Elodi Healy for technical assistance.

## Additional information

### Funding

| Funder | Grant reference number | Author |
| --- | --- | --- |
| National Institute on Drug Abuse | R01DA011289 | Julie A Kauer |
| Brain and Behavior Research Foundation | Young Investigator Award | Abigail M Polter |
| National Institute of Mental Health | K99MH106757 | Abigail M Polter |
| National Institute on Drug Abuse | R01DA040965 | Travis E Brown |
| National Institute of General Medical Sciences | P30 GM 103398-32128 | Travis E Brown |

The funders had no role in study design, data collection and interpretation, or the decision to submit the work for publication.

### Author contributions

AMP, Conceptualization, Formal analysis, Supervision, Funding acquisition, Investigation, Methodology, Writing—original draft, Project administration, Writing—review and editing; KB, RWC, PMD, Formal analysis, Investigation, Writing—review and editing; NMG, Investigation, Writing—review and editing; TEB, Supervision, Funding acquisition, Investigation, Writing—review and editing; JAK, Conceptualization, Formal analysis, Supervision, Funding acquisition, Writing—original draft, Project administration, Writing—review and editing

### Author ORCIDs

Abigail M Polter, http://orcid.org/0000-0003-0151-0996
Julie A Kauer, http://orcid.org/0000-0002-3362-1642

### Ethics

Animal experimentation: All procedures were carried out in accordance with the guidelines of the National Institutes of Health for animal care and use, and were approved by the Brown University Institutional Animal Care and Use Committee ( protocol #1411000106) or by the University of Wyoming Institutional Animal Care and Use Committee (protocol # #20150909TB00195-92).

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
