## [Decision Letter]

Thank you for submitting your article "Constitutive activation of kappa opioid receptors at VTA inhibitory synapses following acute stress" for consideration by *eLife*. Your article has been favorably evaluated by Richard Aldrich (Senior Editor) and three reviewers, one of whom, Lisa Monteggia (Reviewer #1), is a member of our Board of Reviewing Editors.

The reviewers have discussed the reviews with one another and the Reviewing Editor has drafted this decision to help you prepare a revised submission. We hope you will be able to submit the revised version within two months.

Summary:

Each reviewer has identified issues that require clarification. A revised manuscript addressing these issues in the text is needed. However, all reviewers agree that the revisions do not need new data or experiments. Please see the specific comments of the reviewers below.

*Reviewer #1:*

The authors present strong convergent data supporting the conclusion that acute stress triggers constitutive activation kappa opioid receptors independent of dynorphin release. Initial dynorphin dependent activation of kappa receptors later transitions to a "dynorphin-independent" mode and continually impair GABAergic LTP seen in these synapses. Overall, this is a highly competent study, with excellent juxtaposition of in vivo and ex-vivo methods that clearly delineates what happens at these synapses after acute stress. That said, I have some minor questions that need clarification before publication.

1) The constitutive activity of kappa ORs can be regulated by *JNK* signaling but it is stated that this signaling is not involved in the transition to constitutive activity. The authors indicate several possibilities, are there any hints into what this signaling may be? Does this transition involve acute-stress related hormonal mechanisms?

2) How do constitutively active kappa-ORs impact LTPGABA? In the model, they seem to impair GABA release. Is this due to constitutive impairment of presynaptic Ca^2+^+ channels? Is the release machinery affected? Is there a decrease in spontaneous GABA release? Some additional mechanistic insight, which may already be available, would close this gap in the narrative.

*Reviewer #2:*

The manuscript is the next installment in an exciting series of studies from the Kauer lab defining the relationship between stress, inhibitory signaling in putative DA neurons, KORs and stress-induced drug-seeking behavior. The data here provide a strong case for persistent ligand-independent constitutive activation of KORs in stress-induced changes in plasticity at inhibitory synapses onto VTA DA neurons (LTP-GABA). The experiments are straightforward, well controlled and clearly described and the results are robust, meaningful and thoughtfully interpreted.

The only exceptions are related to the LTP-GABA effect and its significance. First, a minor (easily addressable) concern is that although previous papers from the Kauer lab identified and characterized LTP-GABA, there is no mention of the means by which it is induced or how the signaling cascade for induction may be modified by the experimental manipulations until SNAP pops up out of nowhere in the last paragraph of the subsection “Transient κOR activation leads to persistent κOR activity”. This is confusing.

Second, related to this, while the Discussion is generally thorough, there is scant mention of how LTP-GABA, which is a main measure of the study, might be induced in vivo and what its role might be in normal function. While this would of course involve some speculation, given the nature of the model, some discussion seems warranted.

*Reviewer #3:*

This is an interesting study.

Are differences in constitutive KOR activity between GABA and glutamate synapses due to a lack of signaling at excitatory synapses or recruitment of different signaling cascades in the two types of synapses? Experiments determining whether KOR mediated inhibition of GABA and glutamate synapses is mediated by the same or different signaling cascades would be very informative to determine potential explanation for lack of constitutive activity on excitatory synapses after stress. Is differential *JNK* signaling responsible for the lack of effects on excitatory synapses.

Previous work from the Williams lab has shown that U69,593 produces differential inhibitory effects on GABAA and GABAB IPSCs in NAcc and BLA-projecting cells (Ford et al. 2006). Based on the lack of effect by KOR activation on GABAA IPSCs reported in a supplementary figure in the author's initial Neuron paper (Grazienne et al) describing the role of KOR in stress-induced inhibition of GABA LTP, it is possible that recordings may have been biased towards NAcc-projecting DA neurons. For example, the Williams lab found these neurons to lack KOR regulation of GABAA synapses but GABAB synapses.

Do the authors think they are biasing recordings towards this population of cells? Does stress or KOR activation inhibit SNAP-induced LTP of GABAB synapses? Do the observed constitutive effects differ depending on projection target?

It would be interesting to know whether stress alters the ability of a KOR agonist to inhibit GABAA or GABAB IPSCs to determine whether constitutive KOR activity occludes further KOR regulation of inhibitory synapses.

---

## [Author Response]

*Reviewer #1:*

*[…] 1) The constitutive activity of kappa ORs can be regulated by JNK signaling but it is stated that this signaling is not involved in the transition to constitutive activity. The authors indicate several possibilities, are there any hints into what this signaling may be? Does this transition involve acute-stress related hormonal mechanisms?*

There are several possibilities for mechanisms by which constitutive activity of kORs may be triggered. As discussed in the manuscript, orexin/hypocretin is co-released with dynorphin in the VTA, and it is possible that coordinated signaling between these two neuropeptides (or dynorphin and another neurotransmitter) is necessary for constitutive activation of the kOR. Our previous studies (Niehaus et al., EJN 2010; Polter et al., Biological Psychiatry 2014) have shown that glucocorticoid receptors are necessary for the stress-induced blockade of LTPGABA and that this occurs upstream of kOR activation. However, glucocorticoids are only transiently involved in the blockade of LTPGABA. This led us to hypothesize that glucocorticoid actions on dynorphinergic cell bodies in the nucleus accumbens, hypothalamus, or BNST may induce release or expression of dynorphin. It is also possible, however, that glucocorticoid receptors in the VTA may play a role in triggering constitutive activation of kORs. At the moment, this is the state of our knowledge. We have now expanded on this point in the Discussion section.

*2) How do constitutively active kappa-ORs impact LTPGABA? In the model, they seem to impair GABA release. Is this due to constitutive impairment of presynaptic Ca^2+^+ channels? Is the release machinery affected? Is there a decrease in spontaneous GABA release? Some additional mechanistic insight, which may already be available, would close this gap in the narrative.*

We have shown previously that LTPGABA occurs through nitric-oxide mediated activation of guanylyl cyclase in the presynaptic terminal, production of cGMP, and activation of PKG (Nugent et al., Nature 2007, Nugent et al., Neuropsychopharmacology, 2009). Beyond this step, it remains unclear how PKG activation leads to an increase in GABA release. Our previous work (Graziane et al., Neuron, 2013, supplemental Figure 1) showed that there was no change in mIPSC frequency following cold water swim stress, suggesting that changes in basal GABA release are not necessary for the block of LTPGABA by kORs. We therefore favor a mechanism by which constitutively-active kORs act on a substrate that limits induction of plasticity without affecting basal release mechanisms, perhaps through downregulation of soluble guanylyl cyclase, or scaffolding changes that sequester PKG from its (unidentified) substrates.

We have added a discussion of this to the manuscript.

*Reviewer #2:*

*[…] The only exceptions are related to the LTP-GABA effect and its significance. First, a minor (easily addressable) concern is that although previous papers from the Kauer lab identified and characterized LTP-GABA, there is no mention of the means by which it is induced or how the signaling cascade for induction may be modified by the experimental manipulations until SNAP pops up out of nowhere in the last paragraph of the subsection “Transient κOR activation leads to persistent κOR activity”. This is confusing.*

*Second, related to this, while the Discussion is generally thorough, there is scant mention of how LTP-GABA, which is a main measure of the study, might be induced in vivo and what its role might be in normal function. While this would of course involve some speculation, given the nature of the model, some discussion seems warranted.*

We thank the reviewer for noting lack of clarity. We have now added a brief description of the mechanism of LTPGABA to the introduction, and noted earlier in the Results section that LTPGABA can conveniently be induced by SNAP. We have also added discussion of the potential physiological role of LTPGABA.

*Reviewer #3:*

*This is an interesting study.*

*Are differences in constitutive KOR activity between GABA and glutamate synapses due to a lack of signaling at excitatory synapses or recruitment of different signaling cascades in the two types of synapses? Experiments determining whether KOR mediated inhibition of GABA and glutamate synapses is mediated by the same or different signaling cascades would be very informative to determine potential explanation for lack of constitutive activity on excitatory synapses after stress. Is differential JNK signaling responsible for the lack of effects on excitatory synapses.*

Differences in receptor-mediated signaling between kORs at excitatory and inhibitory are the most likely mechanisms underlying the different susceptibility of these receptors to long-term constitutive activity. Given that *JNK* is involved in the rescue of LTPGABA by norBNI but not the actual conversion of the receptor to a constitutively active state, it seems unlikely that differential *JNK* signaling is the culprit, but instead we believe that differential activity of another signaling pathway is required. It is also possible, as mentioned in the Discussion, that conversion of kappa receptors to a constitutively active state requires coordinated signaling with another receptor, such as the orexin receptor, and that this additional receptor is lacking at excitatory synapses. Further work on this point will be valuable, but is somewhat hampered by the presumed presynaptic terminal location of the signaling (inaccessible using pipette-delivered drugs, e.g.). We have added brief mention of these ideas to the Discussion.

*Previous work from the Williams lab has shown that U69,593 produces differential inhibitory effects on GABAA and GABAB IPSCs in NAcc and BLA-projecting cells (Ford et al. 2006). Based on the lack of effect by KOR activation on GABAA IPSCs reported in a supplementary figure in the author's initial Neuron paper (Grazienne et al) describing the role of KOR in stress-induced inhibition of GABA LTP, it is possible that recordings may have been biased towards NAcc-projecting DA neurons. For example, the Williams lab found these neurons to lack KOR regulation of GABAA synapses but GABAB synapses.*

*Do the authors think they are biasing recordings towards this population of cells? Does stress or KOR activation inhibit SNAP-induced LTP of GABAB synapses? Do the observed constitutive effects differ depending on projection target?*

It is likely that our recordings are biased towards the accumbens-projecting dopamine neurons, however; there is a significant amount of conflicting data on electrophysiological and anatomical markers of dopamine neuron subsets. We have now added discussion of the potential subset of dopamine neurons to the manuscript. Regarding GABAB synapses, our previous study (Nugent et al., Neuropsychopharmacology, 2009) showed that SNAP-induced LTPGABA is not present at GABAB synapses, therefore it is not possible to evaluate whether stress or kOR activation blocks plasticity at these synapses. This intriguing finding suggests that different GABAergic terminals have distinct capabilities to support this form of LTP, an area under study in the lab now.

*It would be interesting to know whether stress alters the ability of a KOR agonist to inhibit GABAA or GABAB IPSCs to determine whether constitutive KOR activity occludes further KOR regulation of inhibitory synapses.*

Recent work suggests that GABAB synapses and GABAA synapses onto VTA dopamine neurons arise from different inputs (Edwards et al., Nature Neuro, 2017). We would then expect that while the neurons recorded in this study may have kOR-sensitive GABAB synapses, those synaptic terminals would not be the same as those providing the GABAA input. Therefore, these effects are likely occurring at kORs on different GABAergic inputs, and (as shown in this study), constitutive activity of kORs at one input does not imply constitutive activity at another. Studying this in full will be an exciting future direction for research, but is beyond the scope of the current study. We have added mention of this to our Discussion.